# Refined mapping of tree cover at fine-scale using time-series Planet-NICFI and Sentinel-1 imagery for Southeast Asia (2016-2021)

Feng Yang[a], Zhenzhong Zeng[a, *]

[a] School of Environmental Science and Engineering, Southern University of Science and Technology, Shenzhen 518055, China

[*] Correspondence to: zengzz@sustech.edu.cn (Zhenzhong Zeng)

Mailing Address:

College of Engineering N808

Southern University of Science and Technology

Shenzhen, China

The manuscript for *Earth System Science Data*

August 10, 2023

**Abstract:**

High-resolution mapping of tree cover is indispensable for effectively addressing tropical forest carbon loss, climate warming, biodiversity conservation, and sustainable development. However, the availability of precise high-resolution tree cover map products remains inadequate due to the inherent limitations of mapping techniques utilizing medium-to-coarse resolution satellite imagery, such as Landsat and Sentinel-2 imagery. In this study, we have generated an annual tree cover map product at a resolution of 4.77 m for Southeast Asia (SEA) for the years 2016-2021 by integrating Planet-Norway's International Climate & Forests Initiative (NICFI) imagery and Sentinel-1 Synthetic Aperture Radar data. We have also collected annual tree cover/non-tree cover samples to assess the accuracy of our Planet-NICFI tree cover map product. The results show that our Planet-NICFI tree cover map product during 2016-2021 achieve high accuracy, with an overall accuracy of $\geq 0.867 \pm 0.017$ and a mean F1 score of 0.921, respectively. Furthermore, our tree cover map product exhibits high temporal consistency from 2016 to 2021. Compared to existing map products (FROM-GLC10, ESA WorldCover 2020 and 2021), our tree cover map product exhibits better performance, both statistically and visually. Yet, the imagery obtained from Planet-NICFI performs less in mapping tree cover in areas with diverse vegetation or complex landscapes due to insufficient spectral information. Nevertheless, we highlight the capability of Planet-NICFI imagery in providing quick and fine-scale tree cover mapping to a large extent. The consistent characterization of tree cover dynamics in SEA's tropical forests can be further applied in various disciplines. Our data from 2016 to 2021 at a 4.77 m resolution are publicly available at https://cstr.cn/31253.11.sciencedb.07173 (Yang and Zeng, 2023).

## 1 Introduction

Forests and tree-based systems outside forests play a crucial role in land-based carbon emissions or removals,

making them essential for supporting and monitoring the implementation of the Reducing Emissions from
Deforestation and Forest Degradation (REDD+) and other land-based activities under the Paris Agreement
(Skea et al., 2022; CoP26, 2021; FAO, 2020). However, current forest cover map products exhibit large errors
in accurately estimating forest area and change, particularly in areas such as trees outside forests and forest
edge landscapes (Mugabowindekwe et al., 2023; Reiner et al., 2023; Brandt et al., 2020). As a result, there is
a growing demand for timely, high-quality, and high-resolution tree cover map products to accurately capture
the dynamics and changes in forest cover.

Many tree cover map products have been developed at medium-to-coarse resolutions (10-500 m), such as
Finer Resolution Observation and Monitoring of Global Land Cover 10 m (FROM-GLC10; Gong et al.,
2019), Environmental Systems Research Institute (ESRI) Land Cover (2017-2021) (Karra et al., 2021),
European Space Agency (ESA) WorldCover 2020 and 2021 (Zanaga et al., 2022; Zanaga et al., 2021), GFC
(Hansen et al., 2013), Globeland30 (Chen et al., 2015), Copernicus Global Land Service (CGLS) Land Cover
(Buchhorn et al., 2020), ESA Climate Change Initiative (CCI) (ESA, 2017) and the National Aeronautics and
Space Administration (NASA) MCD12Q1 (Friedl and Sulla-Menashe, 2019). However, accurate high-
resolution tree cover map products at continental-to-global scales are still lacking due to mapping through
medium-to-coarse resolution imagery (Zanaga et al., 2021; Hansen et al., 2010). Consequently, some
uncertainties occur in acquiring global tree inventories and monitoring forest disturbances (deforestation and
forest degradation). This is mainly due to isolated trees or long narrow forest cover removal (Reiner et al.,
2023; Wagner et al., 2023; Sexton et al., 2016; Hammer et al., 2014; Hsieh et al., 2001).

Only recently have two tree cover map products at <4.77 m been produced over Africa and the state of Mato
Grosso in Brazil using Planet-Norway's International Climate & Forests Initiative (NICFI) imagery based on
deep learning algorithms (Reiner et al., 2023; Wagner et al., 2023). However, these two maps have only
limited temporal or spatial coverage that occurred. Since the early 21st century, agricultural expansion has
created a new wave of drastic land use/land cover changes in Southeast Asia (SEA), leading the region to be
one of the most deforested regions worldwide (Zeng et al., 2018a; Zeng et al., 2018b; Achard et al., 2014).
Average elevations and slopes of forest loss sites have significantly increased in SEA, particularly in the
2010s, geometrically irregular upland land use sites commonly occur (Velasco et al., 2022; Feng et al., 2021).
However, existing tree cover map products have underestimated deforestation (25-116%) and upland
agricultural expansion rates (9-113%), especially on the topographic boundaries in SEA (Zeng et al., 2018a).
Thus, fine-resolution tree cover map products in SEA, with high spatial resolution and longer consistent time
series, are urgently needed to accurately monitor tree cover loss and related illegal deforestation. In addition,
combining high-resolution optical imagery and Synthetic Aperture Radar (SAR) data (e.g., Sentinel-1) to
produce large-area tree cover map products is still in its early stage (Zanaga et al., 2022; Karra et al., 2021;
Zanaga et al., 2021; Buchhorn et al., 2020; Hansen et at., 2010).

Concurrently, advances in large-scale cloud computing (e.g., Google Earth Engine, GEE; Gorelick et al.,
2017) and available high-resolution satellite imagery (Roy et al., 2021) can facilitate the development of
high-resolution and longer time-series tree cover map products at continental-to-global scales. In this paper,
we generated a state-of-the-art fine-scale open-source tree cover map product for SEA during 2016-2021
using Planet- NICFI imagery, Sentinel-1 SAR data, and the random forest (RF) method from a previous study
(Yang et al., 2023). This dataset allows for extensive assessments of forest dynamics change, such as
deforestation, forest degradation, and reforestation. In addition, our dataset can monitor trees outside forests
and long narrow forest cover removal, thus improving the accuracy of automated continental tree inventories,
which helps optimize REDD+ under the Paris Agreement.

**2 Materials and methods**
**2.1 Satellite imagery**
We utilized Planet-NICFI and Sentinel-1 imagery for the years 2016-2021 to generate a time series tree cover
map product for SEA. The Planet-NICFI program provides high-resolution (4.77 m per pixel) optical
PlanetScope surface reflectance mosaics specifically designed for the tropics. These mosaics offer accurate
and reliable spatial data with minimized effects from atmosphere and sensor characteristics, making them an
ideal 'ground truth' representation (Planet Team, 2017). The mosaics cover the best imagery to represent every
part of the coverage area during leaf-on periods from June to November based on cloud cover and acutance
(image sharpness). The Planet-NICFI imageries consist of four bands: red, green, blue, and near-infrared, and
cover a time period from 2015 to 2020 at bi-annual resolution for the archive, and from 2020 to 2023 at
monthly resolution for monitoring purposes. We accessed and utilized these products in the GEE platform by
authorizing our NICFI account to the GEE account.

We utilized Sentinel-1 on the GEE platform, specifically the 10 m resolution dual-polarization Ground Range
Detected (GRD) scenes (VV + VH). We chose Sentinel-1 SAR imagery to correct cases of overestimation
caused by confusion with herbaceous vegetation, or underestimation due to optical satellite observations
omitting deciduous or semi-deciduous characteristics (Shimada et al., 2014). The SAR imagery, available
every 12 days for a single satellite or 6 days for a dual-satellite constellation from October 2014 to the present,
was pre-processed with the Sentinel-1 Toolbox for thermal noise removal, radiometric calibration, and terrain
correction.

**2.2 Validation dataset collection**
We collected time series validation datasets to assess the tree cover map product during 2016-2021, except
for 2019 as it has been provided by Yang et al. (2023). Our mapping approach has been comprehensively
assessed after being developed in 2019 (Yang et al., 2023). However, despite the advancements in the Land
Cover Land Use Change (LCLUC) community, a notable gap remains the absence of publicly available high-
resolution (e.g., ≤10 m) tree cover/non-tree cover labels. The existing coarse-resolution labels for tree
cover/non-tree cover can introduce considerable uncertainties when evaluating high-resolution tree cover
maps. As a result, our ability to delve deeper into the accuracy of time-series tree cover map datasets was
hindered.

Following the methodology established by Yang et al. (2023), we undertook a rigorous process to generate a
robust validation dataset for our study. Firstly, we randomly generated 1,515 points to ensure a representative
sample of collected visual data, as illustrated in Fig. 1. Next, to classify these points as trees or non-trees, we
enlisted four human interpreters and employed Planet Explorer within QGIS. Our approach involved visually
identifying tree cover/non-tree cover pixels in the true color composite of Planet-NICFI imagery where the
points were located. To ensure accuracy, we superimposed the 10 m tree height data, previously developed
by Lang et al. (2022), onto the Planet-NICFI imagery. This step ensured that the labels adhered to the specified
tree height criteria (i.e., ≥5 m). Subsequently, we thoroughly evaluated and refined the labels using Google
Earth. To make time series tree cover/non-tree cover labels, we maintained the geographic location of the
1,515 points and changed the year of the Planet-NICFI imagery. The resulting labels encompassed data from
the years 2016, 2017, 2018, 2020, and 2021. Detailed information about the validation dataset can be
presented in Table 1.

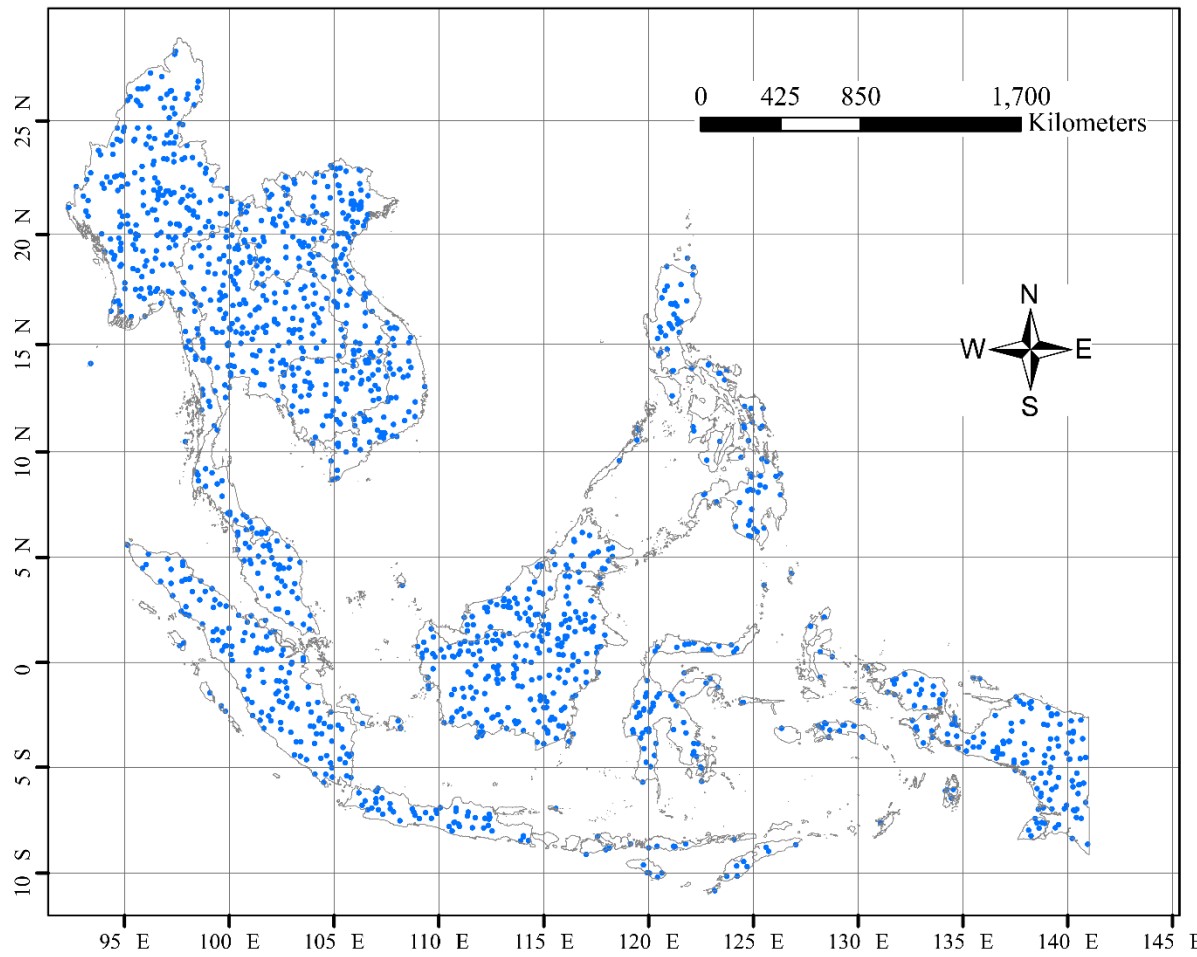


**Figure 1.** Spatial distribution of randomly generated 1,515 validation dataset points.

**Table 1** Information of the mapped validation dataset for evaluating the generated tree cover map product.

| Period | Count of sample points | | |
|--------|------------|----------------|-------|
|        | Tree cover | Non-tree cover | Total |
| 2016   | 1,086      | 429            | 1,515 |
| 2017   | 1,026      | 489            | 1,515 |
| 2018   | 977        | 538            | 1,515 |
| 2020   | 1,093      | 422            | 1,515 |
| 2021   | 952        | 563            | 1,515 |


**2.3 Methods**
We integrated Planet-NICFI and Sentinel-1 SAR imagery to generate a high-resolution (4.77 m) annual tree
cover map product for SEA covering the years 2015-2021. Our framework involved several key steps,
including defining mapped objects, preprocessing of imagery, and generation of time-series tree cover map
product. The detailed workflow is illustrated in Fig. 2.

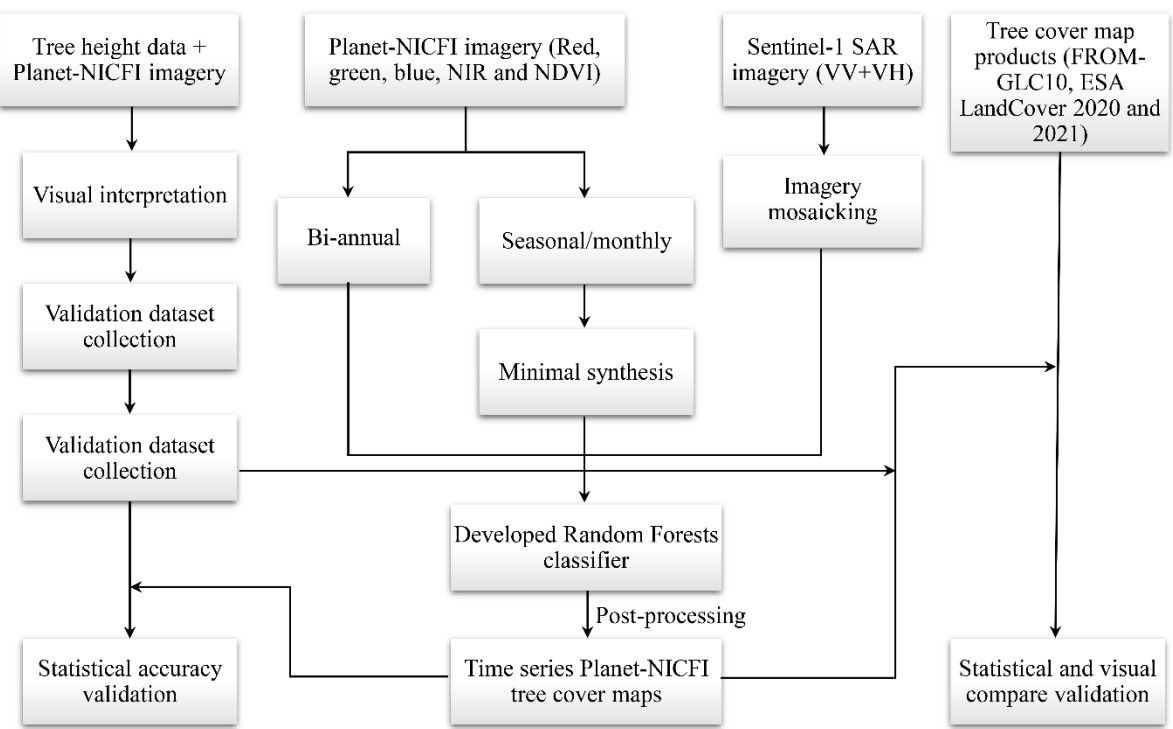


**Figure 2.** Workflow of generating tree cover map product for 2016-2021, including imagery preprocessing,
generation of tree cover map product, and accuracy validation.

2.3.1 Definition of mapped tree cover
Traditionally, forests are considered to meet specific criteria (tree cover and height). The Food and Agriculture
Organization (FAO) of the United Nations defines forests as land spanning more than 0.5 hectares with trees
higher than 5 m and a canopy cover above 10% (FAO, 2020). According to the United Nations Framework
Convention on Climate Change (UNFCCC), forests are defined as areas with a minimum canopy cover of
10-30%, minimum tree height of 2-5 m, and a minimum area of 0.1 ha (Parker et al., 2008).

In this study, tree cover is defined as any geographic area dominated by trees without a percentage of tree
coverage at the pixel level (Zanaga et al., 2020; Hansen et al., 2013). This is attributed to the fact that the
resolution of the Planet pixel (4.77 m) is closer to the size of trees in tropical areas. Next, we utilized Planet-
NICFI imagery to generate only a prototype tree cover map with a resolution of 4.77 m and trees higher than
5 m. Our tree cover map product serves as baseline data for forest cover analysis. Upon further development
of the map to include trees higher than 5/2-5 m, it can be utilized for deriving forest cover maps for various
functions, such as those provided by FAO and UNFCCC.

2.3.2 Preprocessing of imagery
We utilized the GEE platform to preprocess Planet-NICFI imagery and Sentinel-1 SAR data for generating
tree cover maps for the years 2016-2021 (Fig. 2). Specifically, following the methodology of Yang et al.
(2023), we first employed the ee.ImageCollection.mosaic() function to merge and assemble overlapping
Sentinel-1 SAR data over the specified time period into a seamless, continuous imagery. Subsequently, we
performed bilinear resampling on the SAR imagery, specifically the VV and VH bands, to match the spatial
resolution of Planet-NICFI imagery with a spatial resolution of 4.77 m.

Planet-NICFI offers imagery at two different temporal frequencies spanning from 2016 to 2021. This includes
semi-annual imagery from 2016 to 2019 and monthly data from 2020 to 2021. To create a coherent and
consistent dataset for 2020 and 2021, we synthesized the selected time window of monthly imagery into
single imagery for each band, namely red, green, blue, and near-infrared bands. Specifically, we utilized the
ee.ImageCollection.min() function on each monthly imagery to extract the minimum monthly imagery, which
was then used to generate the second semi-annual imagery for 2020 and 2021. This approach was employed
to minimize the impact of cloud pollution on Planet-NICFI imagery (Oishi et al, 2018).

2.3.3 Generation of time-series tree cover map product
In addition to applying the RF approach in our tree cover mapping (Yang et al., 2023), RF-based methods
have been widely employed to develop global LCLUC products and show good performance (Zanaga et al.,
2022; Zanaga et al., 2021; Buchhorn et al., 2020). To acquire the time-series tree cover map dataset, our
methodology involved a two-step process. Initially, we integrated our custom RF approach, implemented on
Google Earth Engine (GEE), with a cloud-based machine learning platform. This combination enabled us to
obtain semi-annual Planet-NICFI and Sentinel-1 imageries spanning the years 2016 to 2021, as illustrated in
Fig. 2. Following data acquisition, we performed several post-processing steps to generate accurate tree cover
map product for the SEA region. These steps included downloading the acquired data from the cloud platform
to a local location, conducting mosaic operations, clipping relevant areas, applying projection transformations,
and performing correlation statistics. By employing this approach, we produced a high-resolution tree cover
map product.

2.3.4 Statistical accuracy assessment
We used two methods to assess the statistical accuracy of our tree cover map product. The generated tree
cover map product was compared pixel by pixel with the tree cover/non-tree cover labels. We then obtained
a confusion matrix, including true tree cover (TP), true non-tree cover (TN), false tree cover (FP), and false
non-tree cover (FN). These four values were used to calculate the user's accuracy, producer's accuracy, and
overall accuracy at a 95% confidence level (Olofsson et al., 2014) and the F1 score based on Eqs. (1)-(4),
respectively. Note that we opted against utilizing the Kappa coefficient for accuracy assessment due to its
unsuitability for mapping error evaluation (Pontius Jr et al., 2011; Allouche et al., 2006).

$$\text{User's accuracy (UA)} = \frac{TP}{TP + FP} \tag{1}$$

$$\text{Producer's accuracy (PA)} = \frac{TP}{TP + FN} \tag{2}$$

$$\text{Overall accuracy} = \frac{TP + TN}{TP + TN + FP + FN} \tag{3}$$

$$\text{F1 score} = \frac{2 \times UA \times PA}{UA + PA} \tag{4}$$


In addition, following Tsendbazar et al. (2021), we used a stability index based on the user's and producer's
accuracy to evaluate the time-series accuracy consistency of the tree cover map product. The stability index
used to evaluate tree cover accuracy is expressed as

$$SI_{t1} = \frac{|TC_{t1} - TC_{t1-1}|}{TC_{t1-1}} \times 100 \tag{5}$$

where $SI_{t1}$ is the stability index that indicates the accuracy of tree cover maps (user's or producer's accuracy)
at time *t1*, $TC_{t1}$ is tree cover accuracy at time *t1* and $TC_{t1-1}$ is tree cover accuracy at the previous time (*t0*
or the reference year). We also used the maximum and average stability index for two consecutive years to
assess the stability of our tree cover map product over a long period.

## 207    3 Results

We employed two approaches to assess the performance of our Planet-NICFI 2016-2021 tree cover map
product. Firstly, we estimated the accuracy of our tree cover map product for each year to gain insights into
their accuracy and consistency, based on the method developed by Tsendbazar et al. (2021). Additionally, we
presented illustrative time series tree cover maps and documented the dynamics in tree cover area changes
during the 2016-2021 period. Secondly, we compared our tree cover map product to widely used global tree
cover map products at 10 m resolution, including FROM-GLC10 in 2017 (Gong et al., 2019), as well as ESA
WorldCover 2020 and 2021 (Zanaga et al., 2022; Zanaga et al., 2021).

**3.1 Assessment of tree cover map product**
We reported the annual accuracy of the time-series Planet-NICFI tree cover map product in Table 2 with a
95% confidence level. The tree cover accuracy results for 2019 were provided by Yang et al. (2023). The
overall accuracy of the tree cover map product ranged between 0.867-0.907 ± 0.015 from 2016 to 2021, with
the highest accuracy of 0.907±0.014 in 2021 and the lowest accuracy of 0.867±0.017 in 2016 (Table 2). This
discrepancy may be due to poor data in the Planet-NICFI imagery during 2016 (Roy et al., 2021). The F1
score showed a similar trend from 2016 to 2021, with an average of approximately 0.921. The user's accuracy
consistently exceeded 0.901±0.017 over the six years, except for 2016 when it was 0.862±0.021. The
producer's accuracies were all higher than 0.912±0.014 (Table 2). Nevertheless, the mapping results of our
time-series Planet-NICFI tree cover maps were highly consistent. Additionally, compared to the tree cover,
the non-tree cover showed lower user's accuracy, producer's accuracy, and F1 score (i.e., approximately
0.856±0.027, 0.852±0.025, and 0.853, respectively), likely due to the complex composition of non-tree cover
types, such as shrubland and herbaceous wetland.

**Table 2** User's accuracies, producer's accuracies, F1 score, and overall accuracies of the Planet-NICFI V1.0
2016-2021 tree cover map product for SEA at a 95% confidence level. The accuracy evaluation results in
2019 were provided by Yang et al. (2023).

| Year | Classification | User's accuracy | Producer's accuracy | F1 score | Overall accuracy |
|------|----------------|-----------------|---------------------|----------|------------------|
| 2016 | Tree cover | 0.862±0.021 | 0.925±0.018 | 0.892 | 0.867±0.017 |
|      | Non-tree cover | 0.876±0.031 | 0.783±0.026 | 0.827 | |
| 2017 | Tree cover | 0.901±0.017 | 0.935±0.016 | 0.917 | 0.892±0.016 |
|      | Non-tree cover | 0.874±0.033 | 0.814±0.027 | 0.843 | |
| 2018 | Tree cover | 0.929±0.016 | 0.912±0.014 | 0.920 | 0.892±0.015 |
|      | Non-tree cover | 0.816±0.033 | 0.85±0.030 | 0.832 | |
| 2019 | Tree cover | 0.913±0.012 | 0.933±0.010 | 0.923 | 0.895±0.011 |
|      | Non-tree cover | 0.857±0.022 | 0.819±0.021 | 0.837 | |

| | | | | | |
|---|---|---|---|---|---|
| 2020 | Tree cover | 0.944±0.014 | 0.927±0.011 | 0.935 | 0.900±0.014 |
| | Non-tree cover | 0.754±0.041 | 0.803±0.040 | 0.778 | |
| 2021 | Tree cover | 0.947±0.014 | 0.934±0.011 | 0.940 | 0.907±0.014 |
| | Non-tree cover | 0.778±0.038 | 0.816±0.039 | 0.796 | |


We also estimated the stability of our Planet-NICFI tree cover maps accuracy over 2016-2021 (Fig. 3). The
results show that the user's and producer's stability indexes were low than 4.5% and 2.5%, respectively,
indicating the good stability of our mapped Planet-NICFI tree cover maps for the six years (2016-2021).

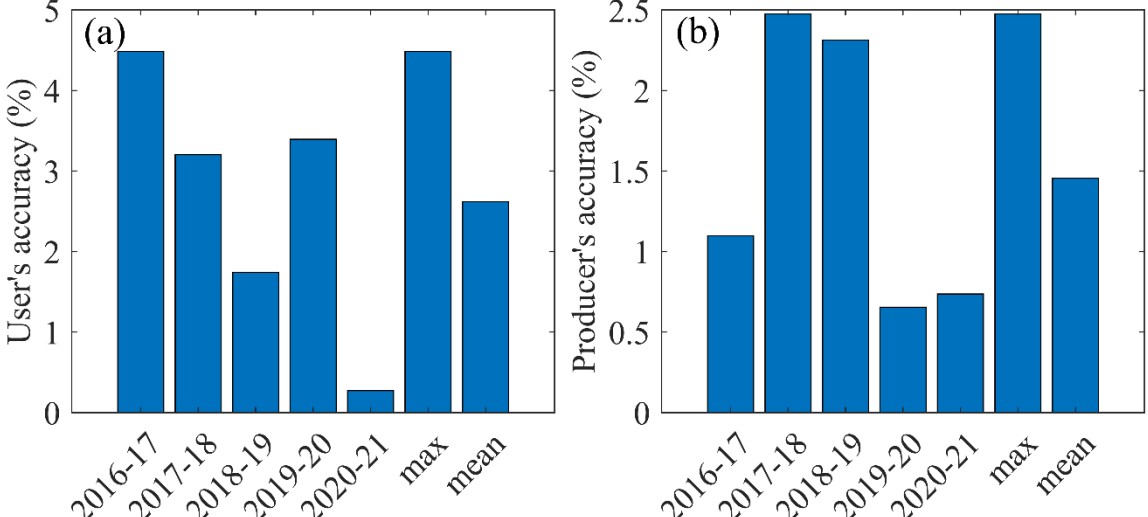

**Figure 3.** Stability index estimates for the Planet-NICFI tree cover map product 2016-2021: the stability
index for (a) the user's accuracy and (b) the producer's accuracy.

We further visually compared our time-series tree cover map product with the original Planet-NICFI imagery
during 2016-2019 (Figures 4-5). Note that we have not shown the years 2020 and 2021 due to inconvenient
visualization for monthly resolution Planet-NICFI imagery collected from QGIS. In comparison, our tree
cover map product showed better consistencies with Planet-NICFI imagery, such as roads, the spatial
distribution pattern of tree cover, and non-tree cover. However, our tree cover product potentially exhibited
a "salt and pepper" phenomenon in some years (i.e., 2017 and 2018) due to the employment of the RF
approach. In practical applications, we need to pay attention to this phenomenon. In addition, we counted the
time series of the area estimates of tree cover maps during 2016-2021 and showed a slight increase trend
from 2016 to 2021, which is in line with the area estimates of ESA tree cover for the years 2020 and 2021.
This may be due to forest restoration after the 2015 El Niño phenomenon (Wigneron et al., 2020), as well as
the impact of expanded plantations (Xu et al., 2020).

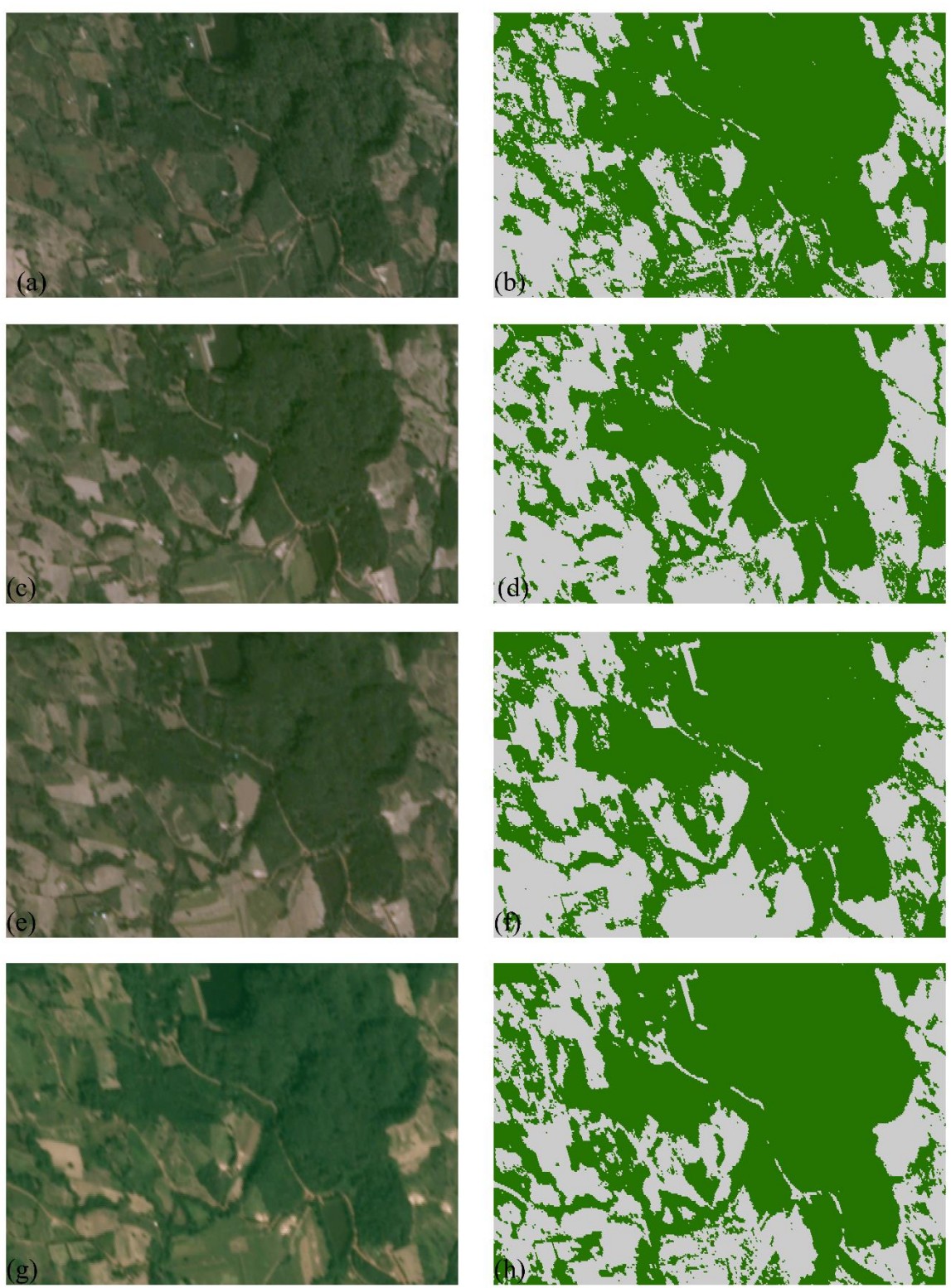

**Figure 4.** Comparison of the time series of the derived tree cover maps (left column) and Planet-NICFI
imagery (right column) for the selected mainland SEA area (100.301°-100.322°E, 18.400°-18.409°N). (a)
and (b), (c) and (d), (e) and (f), and (g) and (h) indicate 2019, 2018, 2017, and 2017, respectively.

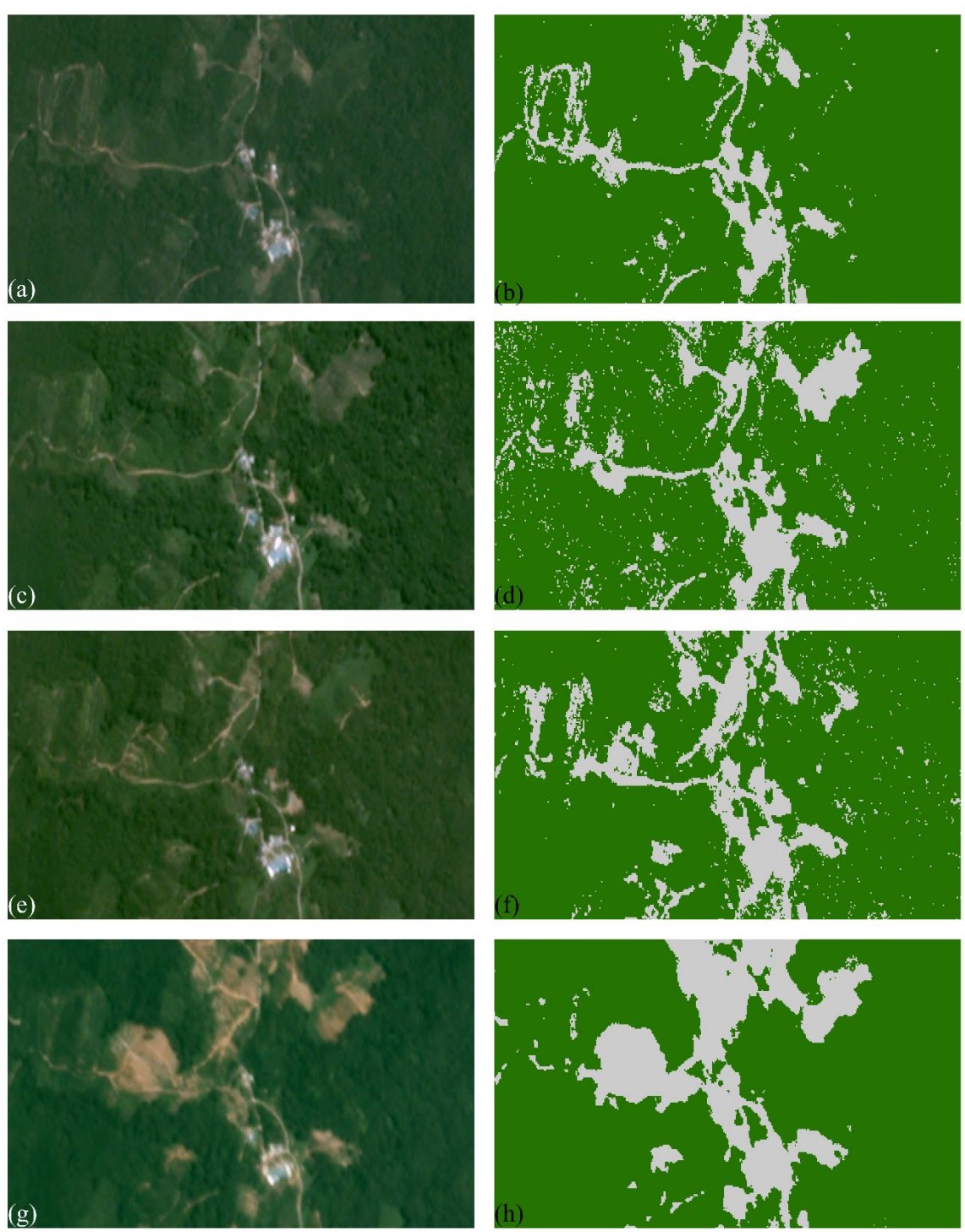

**Figure 5.** Comparison of the time series of the derived tree cover maps (left column) and Planet-NICFI imagery (right column) for the selected maritime SEA area (111.789°-111.806°E, 2.032°-2.040°N). (a) and (b), (c) and (d), (e) and (f), and (g) and (h) indicate 2019, 2018, 2017, and 2017, respectively.

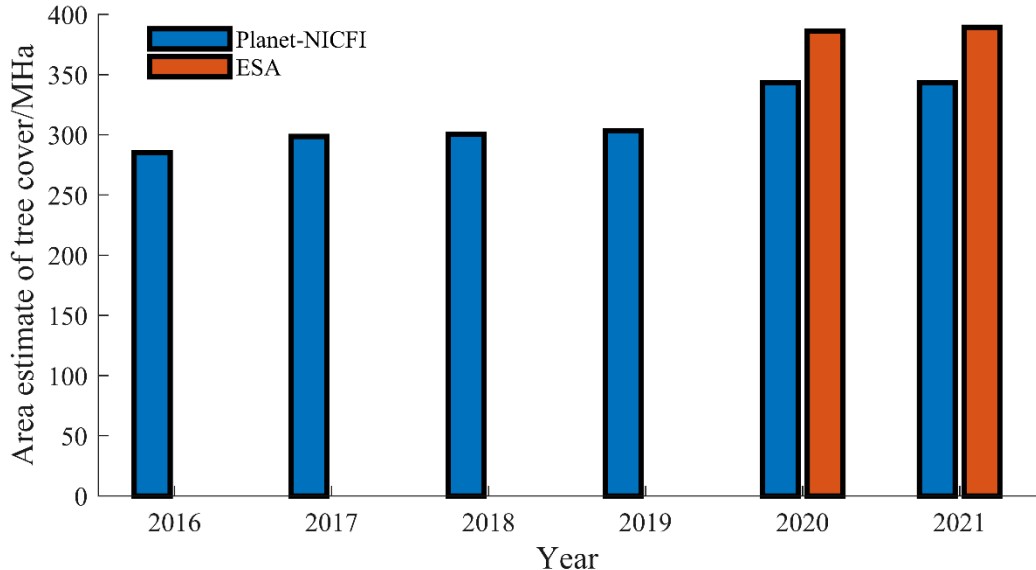

261

**Figure 6.** Area dynamics change of tree cover maps for Planet-NICFI and ESA from 2016 to 2021.

263

**3.2 Comparison with existing tree cover map products**

We compared our mapped Planet-NICFI tree cover maps with FROM-GLC10, ESA WorldCover 2020 and

2021 regarding statistical accuracy (Fig. 4). The results show that our tree cover maps outperformed FROM-

GLC10 in user's accuracy, producer's accuracy, and overall accuracy. The user's accuracy and overall

accuracy of our tree cover maps exceeded 0.083. ESA WorldCover 2020 and 2021 showed similar

performances to our Planet-NICFI tree cover maps. Particularly, the user's accuracy, producer's accuracy,

and overall accuracy of ESA WorldCover 2020 decreased by 0.020, 0.008, and 0.017, respectively (Fig. 4).

This may be because we all used the SAR imagery as input and applied the RF-based machine learning

method to classify our tree cover.

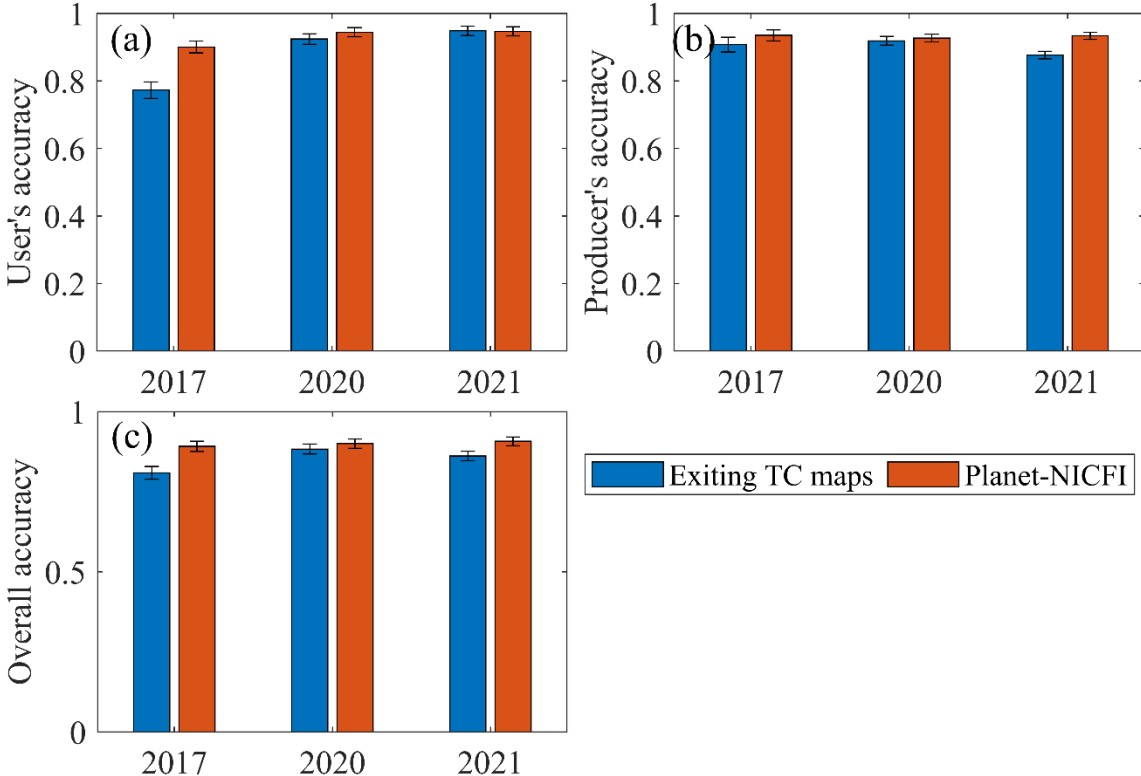

**Figure 7.** Accuracy comparison between existing tree cover maps and the generated Planet-NICFI tree cover maps at a 95% confidence level: (a) user's accuracy, (b) producer's accuracy, and (c) overall accuracy.

We selected six locations (three mainland SEA areas and three maritime SEA areas) to visually compare our Planet-NICFI tree cover maps with three other 10-meter products, namely, FROM-GLC10, ESA WorldCover 2020 and 2021 (Figs. 8-10). In comparison, it is easier for FROM-GLC10 to classify all mixed tree and non-tree areas into non-tree cover maps (Fig. 8a). This may be because FROM-GLC10 cannot apply SAR imagery to tree cover mapping. However, ESA WorldCover 2020 and 2021 can capture tree cover landscapes at a higher level of detail than FROM-GLC, such as long narrow roads, croplands, and built-up areas (Figs. 9-10a). It should be noted that ESA WorldCover 2020 and 2021 omitted some long narrow non-tree cover landscapes and small isolated tree cover and non-tree cover landscapes due to the limitation of the imagery resolution (10 m).

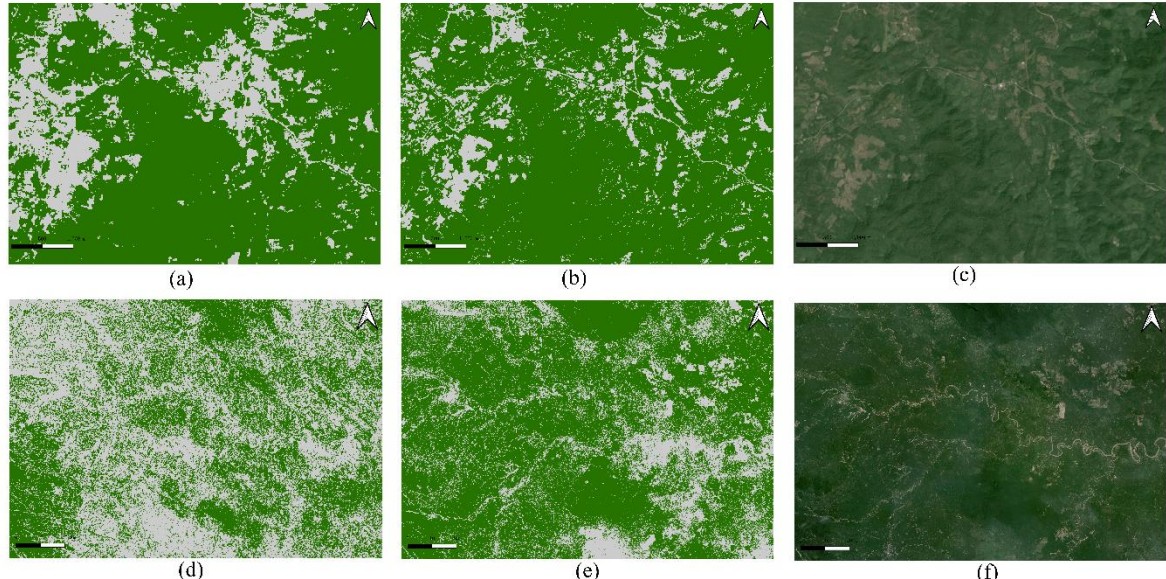

286

**Figure 8.** Comparison of FROM-GLC10 (a) and (d), Planet-NICFI tree cover (b) and (e), and Planet-NICFI imagery (c) and (f) for mainland SEA area (101.594°-101.651°E, 19.254°-19.294°N; top row) and maritime SEA area (101.925°-103.296°E, -2.096°-1.145°S; bottom row). Green and gray 20% indicate tree cover and non-tree cover, respectively.

291

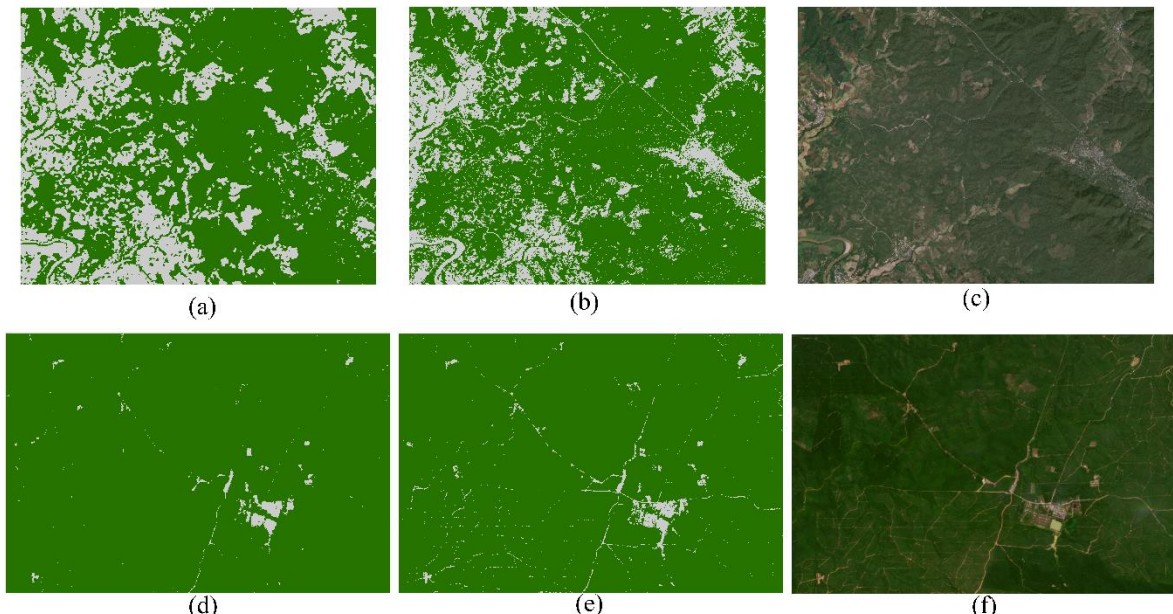

292

**Figure 9.** Comparison of ESA WorldCover 2020 (a) and (d), Planet-NICFI tree cover (b) and (e), and Planet-NICFI imagery (c) and (f) for mainland SEA area (98.310°-98.392°E, 17.102°-17.166°N; top row) and maritime SEA area (99.983°-100.064°E, 1.387°-1.442°N; bottom row). Green and gray 20% indicate tree cover and non-tree cover, respectively.


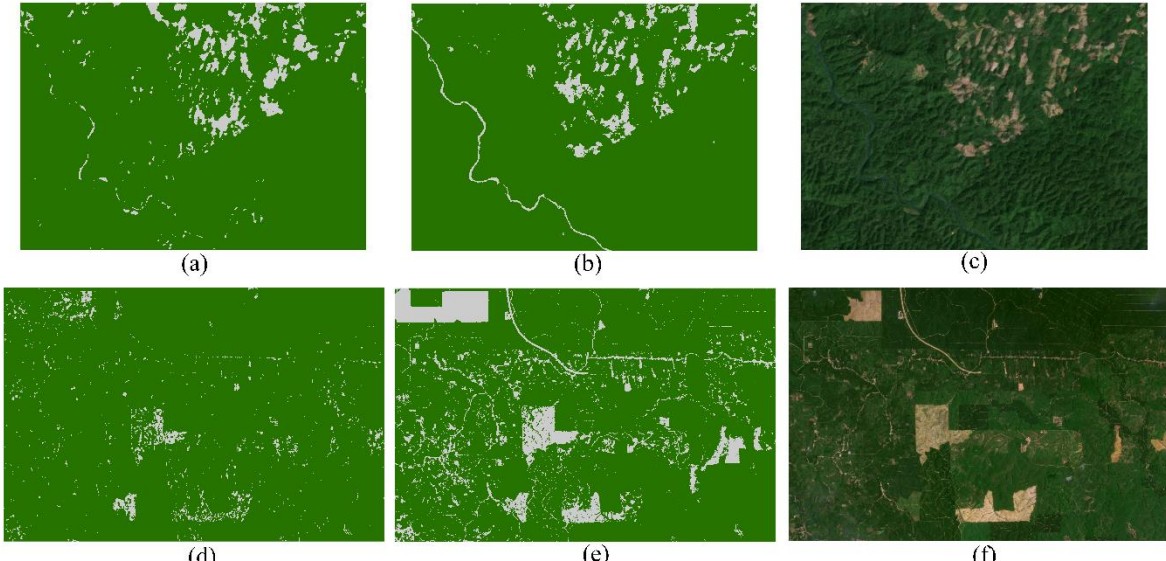


**Figure 10.** Comparison of ESA WorldCover 2021 (a) and (d), Planet-NICFI tree cover (b) and (e), and Planet-
NICFI imagery (c) and (f) for Mainland SEA area (102.179°-102.249°E, 18.676°-18.726°N; top row) and
maritime SEA area (99.951°-100.063°E, 1.892°-1.967°E; bottom row). Green and gray 20% indicate tree
cover and non-tree cover, respectively.

**4 Discussion**

Our time-series Planet-NICFI tree cover map product was mapped twice a year to mitigate the impact of
smog, light, cloud, and topographic effects in tropical areas (Roy et al., 2021; Marta et al., 2018). This high-
resolution tree cover map product meets the minimum tree height requirement of $\geq 5$ m for further generating
forest data. However, it should be noted that we cannot guarantee 100% tree cover for each higher-resolution
pixel, which may introduce some uncertainties when using the higher-resolution tree cover maps. Despite
excluding plantations during sample point labeling, some plantations, such as oil palm, may still be mixed
into our tree cover map product due to similarities in anomalies (Mugabowindekwe et al., 2023; Zanaga et
al., 2022; Zanaga et al., 2021). As a result, caution should be exercised when using our Planet-NICFI tree
cover map product for certain purposes.

To generate a high-resolution time series tree cover map product at a continental scale, we utilized advanced
random forests-based machine learning algorithms on the GEE platform. However, for fine-scale tree cover
mapping, deep learning-based segmentation methods, such as U-net (Falk et al., 2019), are necessary,
particularly when using limited bands (Mugabowindekwe et al., 2023; Wagner et al., 2023; Zanaga et al.,
2022; Zanaga et al., 2021; Brandt et al., 2020). As a result, our tree cover map product still has some
uncertainty due to limitations in the optical PlanetScope imagery. Additionally, our tree cover map product
has the potential to display a salt and pepper phenomenon in certain locations and years, attributed to the
utilization of the RF method. To improve our tree cover mapping product with higher accuracy, we need to
consider adding more bands or utilizing advanced deep learning algorithms in the future.

**5 Data availability**
The high-resolution Planet-NICFI V1.0 time-series tree cover product is now available at
https://cstr.cn/31253.11.sciencedb.07173 (Yang and Zeng, 2023). This product is provided in the Mollweide
projection and the World Geodetic System 1984 (WGS1984) datum and geographic coordinate system. Tree
cover and non-tree cover are denoted as 0 and 1, respectively, in each yearly file, and are stored as UINT8 in
GeoTIFF format. The GeoTIFF files are named Planet-FC_SEA_<YEAR>_prj.tif, for example, Planet-
FC_SEA_16_prj.tif.

**6 Conclusions**
We have successfully generated the first accurate and high-resolution time-series tree cover map product for
SEA by combining optical and SAR satellite observations, utilizing advanced random forests machine
learning algorithms on the GEE platform. Our Planet-NICFI tree cover map product exhibits excellent
accuracy and consistency over six years (2016-2021). The baseline tree cover map product, with a resolution

of 4.77 m, can be easily converted to forest cover maps at different resolutions to cater to the diverse needs

of users. Moreover, our tree cover map product has the unique ability to address rounding errors in forest

cover mapping by accurately capturing isolated trees and monitoring the removal of long, narrow forest cover.

These cutting-edge fine-scale time-series tree cover maps represent a milestone in forest monitoring and offer

unprecedented opportunities for users across diverse disciplines.

**Code Availability**

The scripts used to generate all Planet-NICFI v1.0 tree cover 2016-2021 are provided in JavaScript

(https://code.earthengine.google.com/?scriptPath=users%2Fyftaurus%2Fcodes%3APlanet_RF-LC_rac).

The maps can be automatically generated by running the codes. The scripts are also available on request from

Z. Zeng.

**Acknowledgments**

This study was supported by the National Natural Science Foundation of China (grant no. 42071022), the

start-up fund provided by the Southern University of Science and Technology (no. 29/Y01296122), and the

China Postdoctoral Science Foundation (grant no. 2022M711472). We thank Sen Jiang, Haowen Duan, Hao

Li, and Fangdong Fu for making tree cover/non-tree cover label data that are used to assess the time series

tree cover map products.

**Author contributions**

Z.Z. designed the research; F.Y. performed the analysis and wrote the draft. All authors contributed to the

interpretation of the results and the writing of the paper.

**Competing interests**
The authors declare no competing interests.

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
