# Peer review of "Refined mapping of tree cover at fine-scale using time-series Planet-NICFI and Sentinel-1 imagery for Southeast Asia (2016-2021)"

_Earth System Science Data, 2023_

## Author Comment (AC1)

**Response to the reviewers (#ESSD-2023-143)**

Thanks for the positive comments from the Reviewers. The reviewers' requests are repeated below, in italics, and with our responses written below each suggestion. We have responded in full to each request.

**Reviewer #1 (Remarks to the Author):**

Thanks a lot for your compliments on our paper. We are grateful for your insightful comments and constructive suggestions, which are very helpful for us to improve this manuscript.

Major shortcomings

[*Reviewer #1* **Comment 1**] *The authos claim having produced six annual tree cover maps from 2016 to 2021 based on a random set of 1515 reference samples (visually interpreted) that remain fixed over the 6-year period. However, no example maps are shown that would allow the reader to assess the stability of the derived tree cover over time at the pixel level. The stability of the derived forets cover (at 5m resolution) is also not summarized into statistical numbers - we have to assume that such statistics would show a very large number of implausible forest/non-forest trajectories.*

[**Response**] Thanks a lot for your valuable comments!

In this study, we have generated six annual tree cover maps during 2016-2021 based on the developed machine learning method on the Google Earth Engine platform (Yang et al., 2023). Then, we have made 1515 labels each year to investigate the accuracy of the time series tree cover map product and we find that our product achieves high accuracy, with an overall accuracy of ≥0.867±0.017 and a mean F1 score of 0.921, respectively. Thus, we have successfully generated the first accurate and high-resolution time-series tree cover map product for Southeast Asia by combining optical and SAR satellite observations.

We have added two example time series tree cover maps for the mainland and maritime

Southeast Asia locations from 2016 to 2019, respectively (Figs. R1 and R2), to allow the reader to visually assess our tree cover map product. Note that we have not shown the years 2020 and 2021 due to inconvenient visualization for monthly resolution Planet-NICFI imagery collected from QGIS. Compared to the original Planet-NICFI imagery, our mapped tree cover map products exhibit better accuracy.

In addition, we have counted the time series of the area of tree cover maps during 2016-2021 (Fig. R3) and we showed a slight increase trend for the area of tree cover from 2016 to 2021.

We have added some descriptions in the revised manuscript, "We further visually compared our time-series tree cover map product with the original Planet-NICFI imagery during 2016-2019 (Figures 4-5). Note that we have not shown the years 2020 and 2021 due to inconvenient visualization for monthly resolution Planet-NICFI imagery collected from QGIS. In comparison, our tree cover map product showed better consistencies with Planet-NICFI imagery, such as roads, the spatial distribution pattern of tree cover, and non-tree cover. However, our tree cover product potentially exhibited salt and pepper salt and pepper phenomenon in some years (i.e., 2017 and 2018) due to the employment of the RF approach. In practical applications, we need to pay attention to this phenomenon. In addition, we counted the time series of the area estimates of tree cover maps during 2016-2021 and showed a slight increase trend from 2016 to 2021, which is in line with the area estimates of ESA tree cover for the years 2020 and 2021. This may be due to forest restoration after the 2015 El Niño phenomenon (Wigneron et al., 2020), as well as the impact of expanded plantations (Xu et al., 2020)." (P13L288-P14L298 in the track version of the revised manuscript).

**Reference:**

*Yang, Feng, Xin Jiang, Alan D. Ziegler, Lyndon D. Estes, Jin Wu, Anping Chen, Philippe Ciais, Jie Wu, and Zhenzhong Zeng. Improved fine-scale tropical forest cover mapping for Southeast Asia using Planet-NICFI and Sentinel-1 imagery. Journal of Remote*

*Sensing (2023).*

[Figure]

**Fig. R1** Time series of the derived tree cover maps for the selected mainland Southeast Asia area (100.301°-100.322°E, 18.400°-18.409°N). (a) and (b), (c) and (d), (e) and (f), and (g) and (h) indicate 2019, 2018, 2017 and 2017, respectively.

[Figure]

**Fig. R2** Time series of the derived tree cover maps for the selected maritime Southeast Asia area (111.789°-111.806°E, 2.032°-2.040°N). (a) and (b), (c) and (d), (e) and (f), and (g) and (h) indicate 2019, 2018, 2017 and 2017, respectively.

[Figure]

**Fig. R3** Area dynamics change of tree cover maps for Planet-NICFI and ESA from 2016 to 2021.

[***Reviewer #1* Comment 2**] *The quality of the reference data itself remains unclear and doubtful. In particular, the fixed set of (1515) reference samples shows inter-annual variations that are far from plausible (Tab.1). For example (Tab.1), the samples indicate a 13% tree cover loss between the two consecutive years 2017-2018 followed by a 12% gain the next year (2018 to 2019). The authors do not even mention/discuss this issue - they also fail to indicate possible spill-over effects on the maps produced with this reference data of questionable quality (see above comment).*

[**Response**] Thanks a lot for pointing this out.

We in this study aim to generate a 4.77 m resolution tree cover map product for Southeast Asia during 2016-2021. However, we cannot investigate the accuracy of the time series tree cover map product because the LCLUC community still lacks high-resolution publicly available tree cover/non-tree cover labels. Thus, we conduct strict standards to make the validation labels. Firstly, when the tree height (Lang et al., 2022) is higher than 5 m, we visually identify the tree cover cells in the true color composite of Planet-NICFI imagery. Then, we assess our samples in Google Earth and revised them.

We also have carefully checked the number of tree cover/non-tree cover labels. We find that we miscalculated the numbers for tree cover and non-tree cover labels in 2017 and we have corrected them in Table 1. Please note that this doesn't impact our results. We are very sorry for the misunderstanding caused to your reading due to our carelessness. In addition, a 12% gain from 2018 and 2020 may be reasonable, because our tree cover map product shows a slight increase trend for the area of tree cover from 2016 to 2021, particularly for years 2020 and 2021. Additionally, Planet-NICFI imagery at the monthly resolution collected from QGIS introduces a certain uncertainty in making the tree cover/non-tree cover labels for 2020 and 2021.

We have revised the text in Section 2.2, i.e., "(Yang et al., 2023). However, despite the advancements in the Land Cover Land Use Change (LCLUC) community, a notable gap remains the absence of publicly available high-resolution (e.g., ≤10 m) tree cover/non-tree cover labels. The existing coarse-resolution labels for tree cover/non-tree cover can introduce considerable uncertainties when evaluating high-resolution tree cover maps. As a result, our ability to delve deeper into the accuracy of time-series tree cover map datasets was hindered.

Following the methodology established by Yang et al. (2023), we undertook a rigorous process to generate a robust validation dataset for our study. Firstly, we randomly generated 1,515 points to ensure a representative sample of collected visual data, as illustrated in Fig. 1. Next, to classify these points as trees or non-trees, we enlisted four human interpreters and employed Planet Explorer within QGIS. Our approach involved visually identifying tree cover/non-tree cover pixels in the true color composite of Planet-NICFI imagery where the points were located. To ensure accuracy, we superimposed the 10 m tree height data, previously developed by Lang et al. (2022), onto the Planet-NICFI imagery. This step ensured that the labels adhered to the specified tree height criteria (i.e., ≥5 m). Subsequently, we thoroughly evaluated and refined the labels using Google Earth. To make time series tree cover/non-tree cover labels, we maintained the geographic location of the 1,515 points and changed the year

of the Planet-NICFI imagery. The resulting labels encompassed data from the years 2016, 2017, 2018, 2020, and 2021. Comprehensive information about the validation dataset can be found in Table 1." (P6L116-P7L146 in the track version of the revised manuscript).

**Reference:**

*Lang, Nico, Walter Jetz, Konrad Schindler, and Jan Dirk Wegner. A high-resolution canopy height model of the Earth. arXiv preprint arXiv:2204.08322 (2022).*

[***Reviewer #1* Comment 3**] *The authors claim having combined Planet multi-spectral imagery and S1 (two polarisation) to produce the annual tree cover maps. However, we learn nothing about the respective contribution of the two sensor modalities.*

**[Response]** The points are well taken!

We have given the explanations in the algorithms article (Yang et al., 2023). These mainly include the importance analysis (Fig. R4), as well as the comparison analysis of tree cover maps using Planet/Sentinel-1/Planet only imagery (Fig. R5).

Specifically, the importance analysis shows larger importance values except for the NDVI band, while the comparison analysis finds that introducing additional vegetation structure information can help improve the accuracy of the tree cover map, not mitigate rounding errors. Because we selected the SAR data to address potential overestimation resulting from confusion with herbaceous vegetation, as well as potential underestimation due to optical satellite observations omitting deciduous or semi-deciduous characteristics (Shimada et al., 2014).

**Reference:**

*Shimada, M., Itoh, T., Motooka, T., Watanabe, M., Shiraishi, T., Thapa, R., Lucas, R. New global forest/non-forest maps from ALOS PALSAR data (2007–2010). Remote Sens. Environ., 2014: 155, 13-31.*

[Figure]

**Fig. R4** The importance of the individual band input during the tree cover mapping.

[Figure]

**Fig. R5** Comparing tree cover maps generated using Planet-only and Planet/Sentinel-1 imagery.

Minor comments

[*Reviewer #1* **Specific Comment 1**] *What is labeled as "validation" data in Fig.2 is indeed "training" data.*

[**Response**] No, it is the "validation" data because we in this study aim to generate six annual tree cover maps during 2016-2021 based on the developed machine learning method on the Google Earth Engine platform (Yang et al., 2023), and then investigate the accuracy of the time series tree cover map products by making the tree cover/non-tree cover labels in this study.

[*Reviewer #1* **Specific Comment 2**] *The authors elaborate on the fact that different forest definitions exist (e.g., FAO) but fail to tell the reader which definition was finally adopted. We also learn only in the "Discussion" section, that plantations were excluded from the class "forest" during manual labeling.*

[**Response**] Thanks. We aim to utilize Planet-NICFI imagery to generate a prototype map with a resolution of 4.77 m. Then, our tree cover map products serve as baseline data for forest cover analysis. Upon further development of the map to include trees higher than 5/2-5 m, it can be utilized for deriving forest maps for various functions, such as those provided by FAO and UNFCCC.

[*Reviewer #1* **Specific Comment 3**] *The authors propose a "stability index" (year-to-year change in overall accuracies) "to evaluate tree cover accuracy". Unfortunately, tracking year to year changes in statistical measures will not tell us much about the tree cover accuracy. A good/better plausibility check would have been to compare (pixel-by-pixel) the forest/non-forest trajectories between 2016 and 2021 ... and to analyse if they are at least plausible.*

[**Response**] Thanks. Following Tsendbazar et al. (2021), we mainly leverage the stability index based on the user's and producer's accuracy to investigate the time-series accuracy consistency of the tree cover map products.

In addition, we have added two example time series tree cover maps for the mainland

and maritime Southeast Asia locations from 2016 to 2019, respectively (Figs. R1 and R2), to allow the reader to visually assess our tree cover map product.

**Reference:**

*Tsendbazar, N., Herold, M., Li, L., et al.: Towards operational validation of annual global land cover maps, Remote Sens. Environ., 266, 112686 (2021).*

[**Reviewer #1** **Specific Comment 4**] *Not clear how accuracies are assessed - I guess the authors use the OOB error provided by the RF algorithm?*

[**Response**] Thanks for pointing this out. No, we don't use the OOB error.

We use typical accuracy assessment metrics in the LCLUC community. Specifically, the generated tree cover map products are compared pixel by pixel with the labels. Then, a confusion matrix can be obtained, including true tree cover (TP), true non-tree cover (TN), false tree cover (FP), and false non-tree cover (FN). These four values were used to calculate the accuracy assessment metrics of the draft (Table R1).

**Table R1** Product evaluation metrics and corresponding equations.

| Metric | Equation |
|---|---|
| User's accuracy (UA) | $\dfrac{TP}{TP + FP}$ |
| Producer's accuracy (PA) | $\dfrac{TP}{TP + FN}$ |
| F1-score | $\dfrac{2 \times UA \times RPA}{UA + PA}$ |
| Overall accuracy | $\dfrac{TP + TN}{TP + TN + FP + FN}$ |

We have also added the text in the revised manuscript, which are "product. The generated tree cover map product is compared pixel by pixel with the tree cover/non-tree cover labels. We then obtained a confusion matrix, including true tree cover (TP), true non-tree cover (TN), false tree cover (FP), and false non-tree cover (FN). These four values are used …… based on Eqs. (1)-(4), respectively.

$$\text{User's accuracy (UA)} = \frac{TP}{TP + FP} \tag{1}$$

$$\text{Producer's accuracy (PA)} = \frac{TP}{TP + FN} \tag{2}$$

$$\text{Overall accuracy} = \frac{TP + TN}{TP + TN + FP + FN} \tag{3}$$

$$\text{F1 score} = \frac{2 \times UA \times PA}{UA + PA} \tag{4}$$

" (P10L226-P11L242 in the track version of the revised manuscript).

---

## Author Comment (AC2)

**Response to the reviewers (#ESSD-2023-143)**

Thanks for the positive comments from the Reviewers. The reviewers' requests are repeated below, in italics, and with our responses written below each suggestion. We have responded in full to each request.

**Reviewer #2 (Remarks to the Author):**

[*Reviewer #2* **General Comments**] *This manuscript presents a new method to map distribution of trees between 2016 and 2021 using Planet and Sentinel-1 and RF model, offering high-resolution mapping capabilities. Although this dataset has the potential to make a significant contribution, there are several areas that require improvement, as identified below:*

**[Response]** We sincerely appreciate the reviewer's encouraging words and constructive comments. All issues have been adequately addressed both below and in the revised version of the manuscript.

[*Reviewer #2* **Comment 1**] *1. The manuscript claims to characterize tree cover and its changes, but it only provides a map distinguishing between tree and non-tree, without indicating the percentage of tree coverage. It is essential to reconsider the definition of tree cover in the revised manuscript.*

**[Response]** Thanks a lot for pointing this out.

In this study, tree cover is defined as any geographic area dominated by trees without percentage of tree coverage at the pixel level (Zanaga et al., 2020; Hansen et al., 2013). This is attributed to the fact that the resolution of the Planet pixel (4.77 m) is closer to the size of trees in tropical areas, which leads to some difficulty in seeing tree cover percentage.

We have revised the text in Section 2.3.1 of the revised manuscript, which is "tree cover is defined as any geographic area dominated by trees without percentage of tree coverage at the pixel level (Zanaga et al., 2020; Hansen et al., 2013). This is attributed

to the fact that the resolution of the Planet pixel (4.77 m) is closer to the size of trees in tropical areas." (P9L184-188 in the track version of the revised manuscript).

We have changed the subtitle of Section 2.3.1 to Definition of mapped tree cover.

[*Reviewer #2* **Comment 2**] *2. The resolution of Sentinel-1 images is 10 meters, so it is unclear how the authors obtained tree distribution data at a resolution of 4.77 meters. Further clarification is needed regarding the methodology employed to achieve this higher resolution.*

[**Response**] Thanks a lot for your valuable comment.

The article is an extension of our published algorithms paper (Yang et al., 2023). Thus, this detailed information for changing 10 m Sentinel-1 SAR imagery to a resolution of 4.77 m is given in Section 2.3.2 of the algorithm article. It is "After loading the second biannual Planet red, green, blue, and near-infrared (NIR) bands (i.e., the time window is June to November) and the Sentinel-1 VV/VH bands from 1 June 2019 to 30 November 2019 in the Earth Engine Data Catalog, we leveraged the GEE mosaic function to produce spatially continuous SAR imagery and resampled it using bilinear interpolation to match the spatial resolution of Planet imagery."

We have also revised the text in Section 2.3.2 of the revised manuscript, which is "…match the spatial resolution of Planet-NICFI imagery…" (P9L197-198 in the track version of the revised manuscript).

**Reference:**

*Yang, Feng, Xin Jiang, Alan D. Ziegler, Lyndon D. Estes, Jin Wu, Anping Chen, Philippe Ciais, Jie Wu, and Zhenzhong Zeng. Improved fine-scale tropical forest cover mapping for Southeast Asia using Planet-NICFI and Sentinel-1 imagery. Journal of Remote Sensing (2023).*

[*Reviewer #2* **Comment 3**] *3. While the aim of this dataset is to provide information*

*about changes in tree distribution, the validations conducted thus far seem to focus solely on the spatial pattern of trees. To strengthen the manuscript, it is necessary to include statistical validation regarding changes in tree distribution, ensuring a comprehensive assessment.*

**[Response]** The points are well taken!

We have added two example time series tree cover maps for the mainland and maritime Southeast Asia locations from 2016 to 2019, respectively (Figs. R1 and R2), to allow the reader to visually assess our tree cover map product. Note that we have not shown the years 2020 and 2021 due to inconvenient visualization for monthly resolution Planet-NICFI imagery collected from QGIS. Compared to the original Planet-NICFI imagery, our mapped tree cover map product exhibits better accuracy.

In addition, we have counted the time series of the area of tree cover maps during 2016-2021 (Fig. R3) and we showed a slight increase trend for the area of tree cover from 2016 to 2021.

We have added some descriptions in the revised manuscript, "We further visually compared our time-series tree cover map product with the original Planet-NICFI imagery during 2016-2019 (Figures 4-5). Note that we have not shown the years 2020 and 2021 due to inconvenient visualization for monthly resolution Planet-NICFI imagery collected from QGIS. In comparison, our tree cover map product showed better consistencies with Planet-NICFI imagery, such as roads, the spatial distribution pattern of tree cover, and non-tree cover. However, our tree cover product potentially exhibited salt and pepper salt and pepper phenomenon in some years (i.e., 2017 and 2018) due to the employment of the RF approach. In practical applications, we need to pay attention to this phenomenon. In addition, we counted the time series of the area estimates of tree cover maps during 2016-2021 and showed a slight increase trend from 2016 to 2021, which is in line with the area estimates of ESA tree cover for the years 2020 and 2021. This may be due to forest restoration after the 2015 El Niño phenomenon (Wigneron et al., 2020), as well as the impact of expanded plantations (Xu et al., 2020)." (P13L288-

P14L298 in the track version of the revised manuscript).

[Figure]

**Fig. R1** Time series of the derived tree cover maps for the selected mainland Southeast Asia area (100.301°-100.322°E, 18.400°-18.409°N). (a) and (b), (c) and (d), (e) and (f), and (g) and (h) indicate 2019, 2018, 2017 and 2017, respectively.

[Figure]

**Fig. R2** Time series of the derived tree cover maps for the selected maritime Southeast Asia area (111.789°-111.806°E, 2.032°-2.040°N). (a) and (b), (c) and (d), (e) and (f), and (g) and (h) indicate 2019, 2018, 2017, and 2017, respectively.

[Figure]

**Fig. R3** Area dynamics change of tree cover maps for Planet-NICFI and ESA from 2016 to 2021.

[***Reviewer #2* Comment 4]** *4. It remains unclear how the model performed over complex landscapes and regions with isolated trees. Additional information regarding the model's performance in such settings would greatly enhance the manuscript's credibility and applicability to various environments.*

**[Response]** Thanks a lot for your helpful suggestion.

We have addressed these issues in the algorithms article. Thus, in this study, we mainly report how the data were generated/collected, a thorough evaluation of the time series tree cover map product, the scope and uncertainty of the data, and how to get them, etc.

Specifically, in the algorithms article, we enlarge Planet-NICFI imagery as well as Planet/Sentinel-1 and ESA tree cover maps for four selected focus locations for assessment of built-up fields, dominant croplands in the lowlands, as well as two selected locations in the highlands (Figs. R6-8). We counted conducted the area comparisons of the mapping of new tree cover and ESA LC at six locations (Table R2). In addition, we performed the comparison of pixel-scale fractional cover estimates (Figs. 9-10). Overall, these analyses all show promise for monitoring complex landscapes and regions with isolated trees.

**Table R2** Area comparisons of the mapping new forest cover and ESA LC at six locations.

| Map scene | Elevation/m | Area/km$^2$ | Total forest area/km$^2$ Planet/Sentinel-1 LC | Area difference (%) ESA LC | |
|---|---|---|---|---|---|
| 1 | 182 | 5.8 | 4.5 | 4.6 | 0.1(3.6) |
| 2 | 68 | 136.1 | 123.1 | 128.0 | 4.9(4.0) |
| 3 | 127 | 11.5 | 5.4 | 5.9 | 0.5(9.7) |
| 4 | 36 | 19.1 | 9.0 | 12.4 | 3.4(36.8) |
| 5 | 1050 | 53.7 | 47.4 | 45.3 | −2.1(−4.4) |
| 6 | 845 | 20.0 | 19.3 | 19.9 | 0.6(3.0) |

Note: Area difference indicates ESA LC minus Planet/Sentinel-1 LC, and the values in parenthesis express the area difference as a percentage relative to the area of our forest product.

[Figure]

(a) Planet imagery for dot 1      (b) Planet imagery in dot 2

(c) Planet/Sentinel-1 LC for dot 1      (d) Planet/Sentinel-1 LC for dot 2

(e) ESA LC for dot 1      (f) ESA LC for dot 2

**Fig. R6** Enlarged Planet imagery as well as Planet/Sentinel-1 and ESA forest maps for two selected focus locations for assessment of built-up fields in the lowlands. (a) and (b), (c) and (d), and (e) and (f) indicate Planet imageries, Planet/Sentinel-1 forest map (this study), and ESA forest maps for focus points 1 and 2 in Fig. 4, respectively.

[Figure]

(a) Planet imagery for dot 3

(b) Planet imagery in dot 4

(c) Planet/Sentinel-1 LC for dot 3

(d) Planet/Sentinel-1 LC for dot 4

(e) ESA LC for dot 3

(f) ESA LC for dot 4

**Fig. R7** Same as Figure 6, but for assessment of dominant croplands in the lowlands.

[Figure]

(a) Planet imagery for dot 5

(b) Planet imagery in dot 6

(c) Planet/Sentinel-1 LC for dot 5

(d) Planet/Sentinel-1 LC for dot 6

(e) ESA LC for dot 5

(f) ESA LC for dot 6

**Fig. R8** Same as Figure 6, but for assessment of forest extent in the highlands with elevations above 300 m.

[Figure]

**Fig. R9** Spatial distribution of forest cover fraction maps from Globeland30 in the SEA area (a). The mapped 4.77 m Planet/Sentinel-1 LC product was aggregated into cells of Globeland30 and represented as the forest fraction (percentage) of the cell. (b) zoom in for the selected location of (a).

[Figure]

**Fig. R10** The proportion of forest cover for each map within the interval of 0-0.1, 0.1-0.25, 0.25-0.4, 0.4-0.6, 0.6-0.8, and 0.8-1.

**[*Reviewer #2* Comment 5]** *5. To differentiate this manuscript from the unpublished work of Yang et al. (2023), it is important to highlight the distinguishing features. Providing a clear outline of the unique contributions and methodologies employed in this manuscript compared to Yang et al.'s work will help readers understand the specific advancements and insights presented in each study.*

**[Response]** Thanks a lot for your nice suggestion!

In our algorithms paper, our research question is "the degree to which such high-resolution imagery can mitigate this problem (i.e., "rounding" errors), and thereby improve large-area forest cover maps, is largely unexplored.". In answering this central question, our study achieves two things that we believe are novel. First, we identify one of the main sources of uncertainty in remotely sensed forest cover estimates, which is the "rounding" error resulting from isolated trees and residual tree clumps outside of dense forests that arises when coarser resolution imagery is used to map forests. Second, we demonstrate that using higher resolution PlanetScope imagery can help to minimize such errors, as the potential applications of PlanetScope data for forest cover mapping have not yet been fully explored. Beyond demonstrating the value of PlanetScope for

addressing this issue, we show that PlanetScope on its own does not fully resolve the problem, rather, the best results are achieved by combining high-resolution optical (PlanetScope) and high-medium resolution active remote sensing (Sentinel-1). We explore how this combined methodology addresses the rounding error issue by comparing forest cover estimates derived from our resulting forest cover maps with those from existing forest cover products at ~5 m, 10 m, 25 m, and 30 m resolutions, comparing estimates at the country-level, between mountain and lowland forests, and at the pixel level. We further calculated the proportions of forest cover fraction maps.

In our data description paper, our research question is "the availability of precise high-resolution tree cover map products remains inadequate due to the inherent limitations of mapping techniques utilizing medium-to-coarse resolution satellite imagery, such as Landsat and Sentinel-2 imagery.". Thus, we have generated an annual tree cover map product at a resolution of 4.77 m for Southeast Asia (SEA) for the years 2016-2021 by integrating Planet-Norway's International Climate & Forests Initiative (NICFI) imagery and Sentinel-1 Synthetic Aperture Radar data. We have also collected time-series tree cover/non-tree cover labels to further assess the accuracy of our Planet-NICFI tree cover map products during 2016-2021. Additionally, compared our mapped Planet-NICFI tree cover map with two state-of-art fine-scale tree cover map products (FROM-GLC10, ESA WorldCover 2020 and 2021). Thus, We have successfully generated the first accurate and high-resolution time-series tree cover map product for SEA by combining optical and SAR satellite observations.

We have carefully checked them and differentiated the two papers.

---

## Author Comment (AC3)

**Response to the reviewers (#ESSD-2023-143)**

Thanks for the positive comments from the Reviewers. The reviewers' requests are repeated below, in italics, and with our responses written below each suggestion. We have responded in full to each request.

**Reviewer #3 (Remarks to the Author):**

[*Reviewer #3* **General Comments**] *This manuscript generated an annual tree cover map product at a resolution of 4.77m using Planet and Sentinel-1 data in the period 2016-2021. In general, the research is significant and related-works are well investigated. However, there are still several issues that need to be clarified.*

[**Response**] We sincerely appreciate the reviewer's encouraging words and constructive comments. All issues have been adequately addressed both below and in the revised version of the manuscript.

[*Reviewer #3* **Comment 1**] *1. The author mentions that 1515 validation data are used, but the training data of the RF model is not clearly defined in this manuscript. The manuscript requires explicit definitions of training and validation samples.*

[**Response**] Thanks a lot for pointing this out.

The article is an extension of our published algorithms paper (Yang et al., 2023). The algorithms paper detail introduces definitions and the making of the training labels in Section 2.3.1.

We have also revised the text in Section 2.2 to clarify definitions and the making of the validation samples, i.e., "(Yang et al., 2023). However, despite the advancements in the Land Cover Land Use Change (LCLUC) community, a notable gap remains the absence of publicly available high-resolution (e.g., ≤10 m) tree cover/non-tree cover labels. The existing coarse-resolution labels for tree cover/non-tree cover can introduce considerable uncertainties when evaluating high-resolution tree cover maps. As a result, our ability to delve deeper into the accuracy of time-series tree cover map datasets was

hindered.

Following the methodology established by Yang et al. (2023), we undertook a rigorous process to generate a robust validation dataset for our study. Firstly, we randomly generated 1,515 points to ensure a representative sample of collected visual data, as illustrated in Fig. 1. Next, to classify these points as trees or non-trees, we enlisted four human interpreters and employed Planet Explorer within QGIS. Our approach involved visually identifying tree cover/non-tree cover pixels in the true color composite of Planet-NICFI imagery where the points were located. To ensure accuracy, we superimposed the 10 m tree height data, previously developed by Lang et al. (2022), onto the Planet-NICFI imagery. This step ensured that the labels adhered to the specified tree height criteria (i.e., ≥5 m). Subsequently, we thoroughly evaluated and refined the labels using Google Earth. To make time series tree cover/non-tree cover labels, we maintained the geographic location of the 1,515 points and changed the year of the Planet-NICFI imagery. The resulting labels encompassed data from the years 2016, 2017, 2018, 2020, and 2021. Comprehensive information about the validation dataset can be found in Table 1." (P6L116-P7L146 in the track version of the revised manuscript).

**Reference:**

*Yang, Feng, Xin Jiang, Alan D. Ziegler, Lyndon D. Estes, Jin Wu, Anping Chen, Philippe Ciais, Jie Wu, and Zhenzhong Zeng. Improved fine-scale tropical forest cover mapping for Southeast Asia using Planet-NICFI and Sentinel-1 imagery. Journal of Remote Sensing (2023).*

[***Reviewer #3* Comment 2**] *2. In section 2.3, the author can complement some clear descriptions of how to make comprehensive use of Planet and Sentinel-1 data.*

[**Response**] Thanks a lot for the valuable comment.

We first adjusted Section 2.4 of the previous version to Section 2.3 naming Section 2.3.4. We have then revised the text in Section 2.3 to complement some clear

descriptions of our method, including comprehensive use of Planet and Sentinel-1 data.

These major revised texts are ""To acquire the time-series tree cover map dataset, our methodology involved a two-step process. Initially, we integrated our custom RF approach, implemented on Google Earth Engine (GEE), with a cloud-based machine learning platform. This combination enabled us to obtain semi-annual Planet-NICFI and Sentinel-1 imageries spanning the years 2016 to 2021, as illustrated in Fig. 2. Following data acquisition, we performed several post-processing steps to generate accurate tree cover map product for the SEA region. These steps included downloading the acquired data from the cloud platform to a local location, conducting mosaic operations, clipping relevant areas, applying projection transformations, and performing correlation statistics. By employing this comprehensive approach, we successfully produced a high-resolution tree cover map product." (P10L215-223 in the track version of the revised manuscript).

product. The generated tree cover map product is compared pixel by pixel with the tree cover/non-tree cover labels. We then obtained a confusion matrix, including true tree cover (TP), true non-tree cover (TN), false tree cover (FP), and false non-tree cover (FN). These four values are used …… based on Eqs. (1)-(4), respectively.

$$\text{User's accuracy (UA)} = \frac{TP}{TP + FP} \tag{1}$$

$$\text{Producer's accuracy (PA)} = \frac{TP}{TP + FN} \tag{2}$$

$$\text{Overall accuracy} = \frac{TP + TN}{TP + TN + FP + FN} \tag{3}$$

$$\text{F1 score} = \frac{2 \times UA \times PA}{UA + PA} \tag{4}$$

(P10L226-P11L242 in the track version of the revised manuscript)".

[**Reviewer #3** **Comment 3**] *3. The manuscript has made an evaluation of the forest cover products produced in terms of quantitative assessment and detailed comparison, and the analysis of the results is convincing. However, it lacks a complete display and*

*description of the annual forest cover results in Southeast Asia.*

**[Response]** The point is well taken!

Here we don't show a complete display of time series tree cover maps, mainly because:

- We have showed Southeast Asia tree cover map in 2019 in published algorithms article;
- We can't see detail time series changes in tree distribution when showing these SEA tree cover maps.

Thus, we have added two example time series tree cover maps for the mainland and maritime Southeast Asia locations from 2016 to 2019, respectively (Figs. R1 and R2), to allow the reader to visually assess our tree cover map product. Note that we have not shown the years 2020 and 2021 due to inconvenient visualization for monthly resolution Planet-NICFI imagery collected from QGIS. Compared to the original Planet-NICFI imagery, our mapped tree cover map product exhibits better accuracy. In addition, we have counted the time series of the area of tree cover maps during 2016-2021 (Fig. R3) and we showed a slight increase trend for the area of tree cover from 2016 to 2021.

We have added some descriptions in the revised manuscript, "We further visually compared our time-series tree cover map product with the original Planet-NICFI imagery during 2016-2019 (Figures 4-5). Note that we have not shown the years 2020 and 2021 due to inconvenient visualization for monthly resolution Planet-NICFI imagery collected from QGIS. In comparison, our tree cover map product showed better consistencies with Planet-NICFI imagery, such as roads, the spatial distribution pattern of tree cover, and non-tree cover. However, our tree cover product potentially exhibited salt and pepper salt and pepper phenomenon in some years (i.e., 2017 and 2018) due to the employment of the RF approach. In practical applications, we need to pay attention to this phenomenon. In addition, we counted the time series of the area estimates of tree cover maps during 2016-2021 and showed a slight increase trend from 2016 to 2021, which is in line with the area estimates of ESA tree cover for the years 2020 and 2021.

This may be due to forest restoration after the 2015 El Niño phenomenon (Wigneron et al., 2020), as well as the impact of expanded plantations (Xu et al., 2020(P13L288-P14L298 in the track version of the revised manuscript).

[Figure]

**Fig. R1** Time series of the derived tree cover maps for the selected mainland Southeast

Asia area (100.301°-100.322°E, 18.400°-18.409°N). (a) and (b), (c) and (d), (e) and (f), and (g) and (h) indicate 2019, 2018, 2017, and 2017, respectively.

[Figure]

**Fig. R2** Time series of the derived tree cover maps for the selected maritime Southeast Asia area (111.789°-111.806°E, 2.032°-2.040°N). (a) and (b), (c) and (d), (e) and (f), and (g) and (h) indicate 2019, 2018, 2017, and 2017, respectively.

[Figure]

**Fig. R3** Area dynamics change of tree cover maps for Planet-NICFI and ESA from 2016 to 2021.

---

## Author Comment (AC4)

**Response to the reviewers (#ESSD-2023-143)**

Thanks for the positive comments from the Reviewers. The reviewers' requests are repeated below, in italics, and with our responses written below each suggestion. We have responded in full to each request.

**Reviewer #4 (Remarks to the Author):**

We sincerely appreciate the reviewer's encouraging words and constructive comments. All issues have been adequately addressed both below and in the revised version of the manuscript.

Major questions:

[*Reviewer #4* **Comment 1**] Data validation

*The text says, "We collected other validation datasets to assess the tree cover products during 2016-2021" (Line 111). Does it mean the authors went to the validation regions to check the forest states in person? Or, you are using other datasets or interpretation methods to do validation?*

*In the following paragraph, the authors says, "we randomly generated 1,515 points to ensure the representativeness of collected visual samples". It seems there were field checking. However, in the following sentence "these points were labeled these points as forests or non-forests by four human interpreters using Planet Explorer of QGIS.". Please clarify it.*

[**Response**] Thanks a lot for pointing this out.

Here, we enlisted four human interpreters and employed Planet Explorer within QGIS by interpretation methods to generate a robust annual validation dataset from 2016 to 2021 except for 2019 for investigating the accuracy of the generated time series tree cover map product.

We have revised the text in Section 2.2, i.e., "(Yang et al., 2023). However, despite the

advancements in the Land Cover Land Use Change (LCLUC) community, a notable gap remains the absence of publicly available high-resolution (e.g., ≤10 m) tree cover/non-tree cover labels. The existing coarse-resolution labels for tree cover/non-tree cover can introduce considerable uncertainties when evaluating high-resolution tree cover maps. As a result, our ability to delve deeper into the accuracy of time-series tree cover map datasets was hindered.

Following the methodology established by Yang et al. (2023), we undertook a rigorous process to generate a robust validation dataset for our study. Firstly, we randomly generated 1,515 points to ensure a representative sample of collected visual data, as illustrated in Fig. 1. Next, to classify these points as trees or non-trees, we enlisted four human interpreters and employed Planet Explorer within QGIS. Our approach involved visually identifying tree cover/non-tree cover pixels in the true color composite of Planet-NICFI imagery where the points were located. To ensure accuracy, we superimposed the 10 m tree height data, previously developed by Lang et al. (2022), onto the Planet-NICFI imagery. This step ensured that the labels adhered to the specified tree height criteria (i.e., ≥5 m). Subsequently, we thoroughly evaluated and refined the labels using Google Earth. To make time series tree cover/non-tree cover labels, we maintained the geographic location of the 1,515 points and changed the year of the Planet-NICFI imagery. The resulting labels encompassed data from the years 2016, 2017, 2018, 2020, and 2021. Comprehensive information about the validation dataset can be found in Table 1." (P6L116-P7L146 in the track version of the revised manuscript).

[*Reviewer #4* Comment 2] *In the section "Statistical accuracy assessment", the authors mentioned "the user's accuracy, producer's accuracy, and overall accuracy". Please explain what they are. These terms may be well known in the authors' discipline. However, as a reader of this paper, I don't know what they are, and many readers (and potential users) may have the same issue.*

[Response] Thanks a lot for pointing this out.

We employ widely used accuracy assessment metrics in the LCLUC community to investigate the accuracy of our data. Specifically, the generated tree cover map products are compared pixel by pixel with the labels. Then, a confusion matrix can be obtained, including true tree cover (TP), true non-tree cover (TN), false tree cover (FP), and false non-tree cover (FN). These four values were used to calculate the accuracy assessment metrics of the draft (Table R1).

**Table R1** Product evaluation metrics and corresponding equations.

| Metric | Equation |
|---|---|
| User's accuracy (UA) | $\dfrac{TP}{TP + FP}$ |
| Producer's accuracy (PA) | $\dfrac{TP}{TP + FN}$ |
| F1-score | $\dfrac{2 \times UA \times RPA}{UA + PA}$ |
| Overall accuracy | $\dfrac{TP + TN}{TP + TN + FP + FN}$ |

We have also added the text in Section 2.3.4 of the revised manuscript, which is "product. The generated tree cover maps product is compared pixel by pixel with the tree cover/non-tree cover labels. We then obtained a confusion matrix, including true tree cover (TP), true non-tree cover (TN), false tree cover (FP), and false non-tree cover (FN). These four values are used …… based on Eqs. (1)-(4), respectively.

$$\text{User's accuracy (UA)} = \frac{TP}{TP + FP} \tag{1}$$

$$\text{Producer's accuracy (PA)} = \frac{TP}{TP + FN} \tag{2}$$

$$\text{Overall accuracy} = \frac{TP + TN}{TP + TN + FP + FN} \tag{3}$$

$$\text{F1 score} = \frac{2 \times UA \times PA}{UA + PA} \tag{4}$$

" (P10L226-P11L242 in the track version of the revised manuscript).

[***Reviewer #4* Comment 3**] *Section "6 Conclusions"*

*For a data paper, we have read the abstract, and understand how the data were generated/collected, the scope and uncertainty of the data, and know how to get them.*

*Do we really need a "conclusion" section?*

**[Response]** The points are well taken!

In our study, we make use of published data description papers that contain distinct components, such as "abstract" and "conclusion." To ensure clarity and differentiation between these sections, we adopt specific approaches for each.

In the abstract, we introduce the background and research question as well as how the data were generated/collected, the scope and uncertainty of the data, and how to get them, etc.

On the other hand, the "conclusion" section offers a deeper analysis of the implications of our data. Here, we go beyond the introductory aspects mentioned in the abstract and delve into the broader significance and potential applications of our findings. By emphasizing the implications of our data, we aim to provide a comprehensive understanding of its relevance and impact in the context of the research question and beyond.

Minor questions and edits:

**[*Reviewer #1* Specific Comment 1]** *Line 28: Please clarify what "annual samples" are.*

**[Response]** Thanks. We have changed annual samples to annual tree cover/non-tree cover samples.

**[*Reviewer #1* Specific Comment 2]** *Lines 29~30 "with an overall accuracy of 0.867±0.017 and a mean F1 score of 0.921, respectively." Please explain what "overall accuracy and F1 score" are. My opinion, either explain it clearly or don't mention it. In the previous sentence, authors have said "high accuracy". Since I don't know "overall accuracy and F1 score", it is still "high accuracy" to me.*

**[Response]** Thanks. We reported high accuracy of our tree cover map product measured by overall accuracy and F1 score.

**[*Reviewer #1* Specific Comment 4]** *Lines 31~34 "Compared to existing maps …". These sentences can be removed. Add more details about your data published with this paper.*

**[Response]** Thanks. It is Okay to keep it.

**[*Reviewer #1* Specific Comment 4]** *Lines 37 "The annual Planet-NICFI V1.0 tree cover map products from 2016 to 2021 at 4.77 m resolution". I would replace it with "Our data". It is not necessary to repeat the same information in the abstract.*

**[Response]** Thanks. Have revised it.

**[*Reviewer #1* Specific Comment 5]** *Lines 112~115. Please rephrase this section so we can understand why "except 2019".*

**[Response]** Thanks. We have added the descriptions "as it has been provided by Yang et al. (2023)." (P6L115 in the track version of the revised manuscript).

**[*Reviewer #1* Specific Comment 6]** *Line 118: remove "these points". repeated.*

**[Response]** Thanks. We have removed it.

**[*Reviewer #1* Specific Comment 7]** *Line 119 "four human interpreters": Do you mean you asked four collogues (Homo sapiens) to do a test of identification? Are they acknowledged?*

**[Response]** No, we enlisted four human interpreters and have acknowledged them in the revised manuscript.

**[*Reviewer #1* Specific Comment 8]** *Line 139: remove "For example,".*

**[Response]** Thanks. We have removed it.

**[*Reviewer #1* Specific Comment 9]** *Line 176. Please explain what are "user's accuracy, producer's accuracy, and overall accuracy".*

**[Response]** Thanks. We have added four equations to explain them, which are expressed as

$$\text{User's accuracy (UA)} = \frac{TP}{TP + FP} \tag{1}$$

$$\text{Producer's accuracy (PA)} = \frac{TP}{TP + FN} \tag{2}$$

$$\text{Overall accuracy} = \frac{TP + TN}{TP + TN + FP + FN} \tag{3}$$

$$\text{F1 score} = \frac{2 \times UA \times PA}{UA + PA} \tag{4}$$

We have added them to Section 2.3.4 and also added the text to describe them in the revised manuscript. Please see the response of [*Reviewer #4* Comment 2].

**[*Reviewer #1* Specific Comment 10]** *Lines 186~191: I cannot get what the first approach is.*

**[Response]** Thanks a lot. The first approach is four metrics (user's accuracy, producer's accuracy, and overall accuracy). They are the most commonly used metrics for evaluating data generated.

**[*Reviewer #1* Specific Comment 11]** *Line 188 "based on a study by Tsendbazar":* *Please explain what it is.*

**[Response]** Thanks. We have added the words "the methods developed" (P11L253 in the track version of the revised manuscript).

**[*Reviewer #1* Specific Comment 12]** *Line 195 "The results for 2019 were provided by .." can be move to the method section*

**[Response]** Thanks a lot. We have changed "The results for 2019 were provided by …"

to "The tree cover accuracy results for 2019 were provided by...".

**[*Reviewer #1* Specific Comment 13]** *Line 261 "minimum tree height …": is this about the definition of forest?*

**[Response]** Yes, we have added the words "…for further generating forest data" (P19L362-363 in the track version of the revised manuscript).

**[*Reviewer #1* Specific Comment 14]** *Lines 268~269. I think this sentence is about algorithm "random forest". Please rephrase this sentence.*

**[Response]** Thanks. We have added word "based" after random forests.

**[*Reviewer #1* Specific Comment 15]** *Line 270; "U-net"?*

**[Response]** Thanks. U-Net is a convolutional neural network that was developed for biomedical image segmentation at the Computer Science Department of the University of Freiburg.

We have also added a reference ((Falk et al., 2019)) after the "U-net".

**Reference:**

*Falk, T., Mai, D., Bensch, R., et al. U-Net: deep learning for cell counting, detection, and morphometry. Nature methods, 16(1), pp.67-70, 2019.*

**[*Reviewer #1* Specific Comment 16]** *Line 301 Section "Acknowledgements" who are those four "human interpreters" mentioned in the main text?*

**[Response]** Thanks. We added them to "acknowledgments", i.e., "We thank Sen Jiang, Haowen Duan, Hao Li, and Fangdong Fu for making tree cover/non-tree cover label data that are used to assess the time series tree cover map products." " (P21L408-410 in the track version of the revised manuscript).

---

## Author Response (AR2)

**Uploaded files validated by Lorena Grabowski**

Notification to the authors:

**[Specific Comment]** We noticed that your figures 4 and 5 contain aerial images. If you are not the originator of these images, then appropriate credit or copyright must be given in the figure itself or in the figure caption. If applicable, please adjust this with the next file upload.

**[Response]** Thanks a lot for pointing this out. We have revised the Figure captions of Figures 4 and 5 of the revised manuscript to give the appropriate credit or copyright.

**Public justification (visible to the public if the article is accepted and published):**

Thanks a lot for your compliments on our paper. We are grateful for your valuable comments and suggestions, which are very helpful for us to improve this manuscript.

**[Topical Editor # Comment 1]** *In the realm of accuracy assessment in remote sensing, two widely employed metrics are accuracy and the Kappa coefficient. These metrics play a pivotal role in evaluating the precision and dependability of remote sensing classifications or analyses. While both metrics are significant, it would be beneficial for the authors to elaborate on their choice to omit the Kappa coefficient in the present study. A justification for this decision would provide readers with a clearer understanding of how the chosen metrics align with the study's specific objectives and context.*

**[Response]** Thanks a lot for pointing this out. Given that prior research indicates the inadequacy of the Kappa coefficient for map error assessment (Pontius Jr et al., 2011; Allouche et al., 2006), we have chosen to substitute it with the F1 metric.

We have also added the descriptions in the revised manuscript, i.e., "Note that we opted against utilizing the Kappa coefficient for accuracy assessment due to its unsuitability for mapping error evaluation (Pontius Jr et al., 2011; Allouche et al., 2006)." (P10L202-P11L205 in the track version of the revised manuscript).

**Reference:**

*Allouche, O., Tsoar, A., Kadmon, R.: Assessing the accuracy of species distribution models: prevalence, kappa and the true skill statistic (TSS), J. Appl. Ecol., 43(6), 1223-1232, 2006.*
*Pontius Jr, R. G., Millones, M.: Death to Kappa: birth of quantity disagreement and allocation disagreement for accuracy assessment, Int. J. Remote Sens., 32(15), 4407-4429, 2011.*

**[Topical Editor # Comment 2]** *I've observed a recurring use of the term 'comprehensive' throughout the revised paper. It's worth noting that this term can be highly subjective and should be employed judiciously to ensure its meaningful impact in the manuscript.*

**[Response]** Thanks a lot for your valuable comment.

We have diligently reviewed and refined this aspect in the revised manuscript. Additionally, we meticulously examined other elements of the original text to enhance the overall quality of the article.

---

## Author Response (AR3)

**Remarks from the preceding review file validation**

Notification to the authors:

**[Specific Comment]** *We noticed that your figures 4 and 5 contain aerial images. If you are not the originator of these images, then appropriate credit or copyright must be given in the figure itself or in the figure caption. If applicable, please adjust this with the next file upload.*

**[Response]** Thanks a lot for pointing this out. No, Figures 4 and 5 don't contain aerial images, and they are Planet-NICFI imagery. Additionally, Through Norway's International Climate & Forests Initiative (NICFI), users can now access Planet's high-resolution, analysis-ready mosaics of the world's tropics in order to help reduce and reverse the loss of tropical forests, combat climate change, conserve biodiversity, and facilitate sustainable development for noncommercial uses. We also have revised the Figure captions of Figures 4 and 5 in the previous manuscript.